# From Forgery to Authenticity: Image Anti-Forensics via Reconstruction and Artefact Elimination

## Abstract

In recent years, the development of large-scale vision-language models has resulted in significant advancements in image generation and editing, producing results that can often deceive the naked eye. However, despite their convincing appearance, these generated images remain susceptible to detection by forgery detectors due to various artefacts. The goal of image anti-forensics is to eliminate such artefacts, ensuring that manipulated images successfully evade detection and enhance their overall quality. Existing image anti-forensics methods primarily focus on rectifying artefacts at the feature level, often overlooking the authenticity of the manipulated regions. To address this limitation, we propose a two-phase approach. In the first phase, we introduce GUIded Diffusive rEfinement (GUIDE), a zero-shot learning-based image refinement module aimed at reconstructing details from unaltered regions. In the second phase, we introduce an artefact removal algorithm to eliminate artefacts from the reconstructed "forged regions". We validate the effectiveness of our proposed method across multiple image forgery datasets, and comprehensive ablation studies further affirm the efficacy of each component of our approach. The code will be made available upon acceptance.

## 1 Introduction

As digital media continues to evolve, image manipulation has become increasingly prevalent, offering creative possibilities while simultaneously introducing critical risks to information integrity and public trust (Ahmad & Khursheed (2021), Singh & Kumar (2024)). Common manipulation techniques, including splicing (Kumari & Garg (2024)), copy-move (Abd Warif et al. (2016)), and in-painting (Quan et al. (2024)), often leave detectable traces, particularly in the form of high-frequency artefacts that emerge due to inconsistencies in texture or visual features (Mejri et al. (2021), Wang et al. (2022), Zeng & Pun (2024)). The primary goal of image forensics is to detect such manipulations by identifying anomalies and tracing the altered regions within an image. Conversely, the field of image anti-forensics has developed as a countermeasure, aiming to conceal these traces and improve the visual quality of tampered images, thereby challenging the capabilities of forensic detectors.

Despite the advancements in image anti-forensics, the task remains inherently difficult. Manipulated regions frequently lack any natural correlation with the original content, differentiating this task from conventional image optimisation problems. Conventional methods relying on explicit mathematical models, such as those used for degradation restoration, are often inadequate (Kawar et al. (2022), Li et al. (2022), Yue et al. (2024)). As a result, recent efforts have shifted towards generating realistic details at the feature level and removing forgery traces using generative models, particularly Generative Adversarial Networks (GANs) and trace modelling techniques (Chen et al. (2020), Wesselkamp et al. (2022)). However, these methods typically focus on specific trace types or artefacts, limiting their effectiveness due to dependencies on the underlying forensic models.

Recently, diffusion models have gained significant attraction in image refinement tasks, offering an alternative approach for generating high-quality details in manipulated or degraded images. Notably, methods such as DR2 (Wang et al. (2023)) and DDRM (Kawar et al. (2022)) have successfully ap-

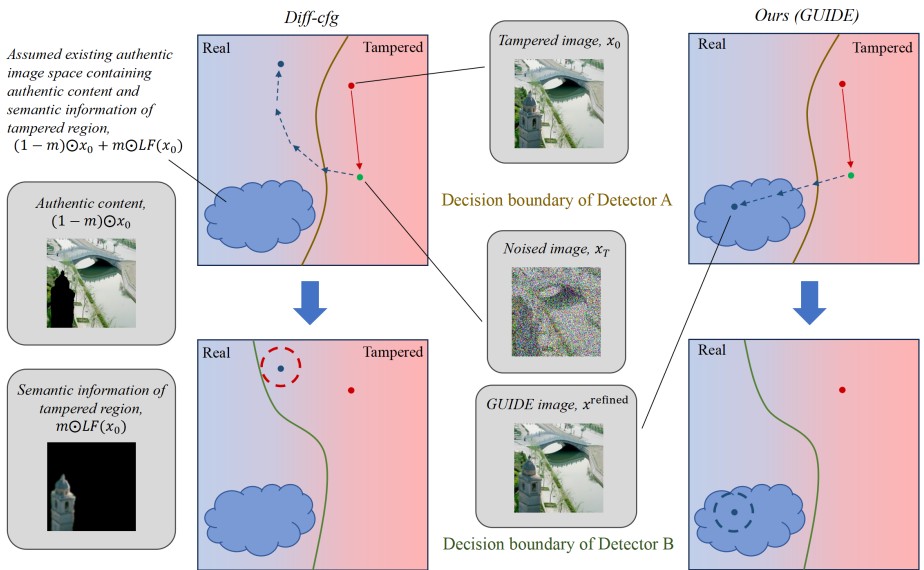

Figure 1: Illustrative representation of the diffusion-based anti-forensics method. We hypothesise the existence of a shared image space that encompasses the authentic regions of a tampered image, which forensic detectors would classify as "real". The blue dashed line represents the stepwise diffusion process. In *Diff-cfg*, a guidance term is employed during the diffusion process, which gradually pulls the refined image back towards the tampered image space, limiting its effectiveness against more robust detectors. In contrast, GUIDE uses only the authentic regions of the tampered image as guidance, improving its ability to evade detection.

plied diffusion models to super-resolution and degradation recovery tasks, underscoring the potential of this approach. Diffusion models operate by learning a denoising process, mapping images—both real and manipulated—into a noise domain before iteratively refining them towards their original state. This process provides a unique opportunity to improve anti-forensic methods by directing the denoising process towards a space that represents real, unmanipulated images.

Inspired by these advances, Tailanián et al. (2024) pioneered the use of guided diffusion models in image anti-forensics with their *Diff-cfg* model, which effectively balances trace removal with content preservation through a guided denoising process. Building upon this foundation, we propose a novel two-stage image anti-forensics framework that leverages guided diffusion refinement to address the limitations of existing methods.

In the first stage of our approach, we introduce GUIded Diffusive rEfinement (GUIDE), a zero-shot diffusion model that uses low-frequency information from tampered regions to eliminate high-frequency artefacts. By incorporating the unique features of authentic regions, this model enhances the overall realism of the manipulated image. As shown in Fig. 1, the motivation behind this work lies in the existence of a real image space, which encompasses the authentic content of a tampered image and the low-frequency components of the semantic information in the tampered regions. In other words, we "reconstruct" the degraded details from this real image space. In the second stage, we propose a texture refinement module to further smooth the visual output and remove residual artefacts. Unlike prior approaches, which are often biased towards the manipulated content, our method projects the denoising process directly into a hypothesised real image space, improving the model's ability to generalise across diverse forensic detectors.

Our contributions are summarised as follows:

- We introduce a zero-shot diffusion-based refinement method that fully exploits the information from authentic regions of tampered images.
- We propose a two-stage refinement framework that achieves state-of-the-art performance across various forensic detection benchmarks.

- We conduct an in-depth exploration of the trade-off between effective image anti-forensics and overall image quality.

## 2 RELATED WORK

In this section, we first provide an overview of two fields closely related to our work: image forensics and image anti-forensics. We then conclude with a brief introduction to the backbone of our method: Denoising Diffusion Probabilistic Models (DDPM).

### 2.1 IMAGE FORENSICS

Image forensics focuses on designing frameworks to effectively detect manipulated images, which primarily involve splicing, copy-move, and inpainting techniques. Common characteristics of tampered images utilised in image forensics include RGB values, noise patterns, and frequency artefacts (Wang et al. (2022)).

Conventional image forensics approaches revolve around feature modelling. Some methods aim to handcraft specific features using advanced deep learning techniques. For instance, Cozzolino & Verdoliva (2020) applied a CNN architecture to extract a noiseprint specific to the camera type. Other methods treat forgery traces as learnable, black-box features. Bappy et al. (2019) employed an LSTM backbone to analyse the relationship between manipulated and authentic blocks within an image. Similarly, Wu et al. (2019) divided the image forensics task into two stages: a forgery trace extractor and a local anomaly detector, which significantly improved detection performance.

Recent advancements in image forensics have shifted towards multi-modal forgery trace identification, presenting new challenges for developing generalisable anti-forensics techniques. For example, TruFor (Guillaro et al. (2023)) introduced an innovative encoder-decoder architecture that fuses RGB and Noiseprint++ modalities, enabling effective detection of image manipulation. Triaridis & Mezaris (2024) further extended this multi-modality fusion approach by incorporating SRM filters and BayarConv2D for feature extraction.

High-frequency traces have also garnered significant attention. Several studies have leveraged high-frequency noise for image forensics. Li & Huang (2019) utilised HPFCN, a high-pass fully convolutional network, to detect deep inpainted regions by extracting image residuals with a high-pass filter and exposing inpainting artefacts. Additionally, Liu et al. (2024) combined high-frequency traces with semantic information to accurately localise tampered regions.

### 2.2 IMAGE ANTI-FORENSICS

Image anti-forensics methods aim to deceive image forensics by removing identifiable traces left after manipulation.

Earlier anti-forensics approaches focus on task-specific feature extraction and corresponding refinement (Böhme & Kirchner (2012)). Conventional techniques typically involve forgery trace suppression and the addition of authentic image traces. The former destroys detectable structures, while the latter restores or synthesises authentic features. For example, Stamm & Liu (2008) utilised a combination of a median filter and additive Gaussian noise to deceive image-camera classifiers, exemplifying structure destruction techniques. In terms of adding authentic traces, Tahir & Bal (2024) revisited these methods, confirming that they still performed effectively against state-of-the-art image forensics techniques.

The advent of deep learning introduced more effective anti-forensics methods, incorporating learnable features. Contemporary anti-forensics research primarily focuses on refining GAN-generated images, as these methods provide more challenging adversarial samples for image generators. Distribution-based attacks have shown promising results by minimising the distance between the manipulated and authentic image spaces. Hou et al. (2023) highlighted that GAN images leave statistical and frequency traces, proposing StatAttack, which applies adversarial blur, noise, and exposure adjustments while using an MMD loss propagation module to reduce the distributional differences between GAN and real images. Wu et al. (2024) separated high-frequency and low-frequency components, refining the low-frequency regions with blurring and the high-frequency

regions with universal imitation attacks, effectively masking residual traces. Studies like Chen et al. (2020) have also focused on specific trace types, such as camera traces, and developed loss functions to minimise them.

Although the aforementioned methods have shown promising results in image anti-forensics tasks, they often fall short of deceiving the human eye, even when they successfully evade forensic detection. Therefore, in this paper, we focus on enhancing the perceived "authenticity" of tampered regions—ensuring that visual discrepancies are imperceptible to the human eye—while effectively removing forensic artefacts from the image.

### 2.3 DIFFUSION

Diffusion has shown impressive capabilities in a wide range of image restoration tasks, including image super-resolution (Wang et al. (2024)), deraining (Wei et al. (2023)) and inpainting (Corneanu et al. (2024)). DR2 (Wang et al. (2023)) uses ILVR-style (Choi et al. (2021)) conditional denoising to generate super-resolution face images. Zheng et al. (2024) proposes Self-Adaptive Reality-Guided Diffusion to iteratively sample images during the latent diffusion process, employing low-resolution ground truth as realistic guidance to remove perceptual artefacts.

DDPM (Ho et al. (2020)) is basically an u-net model that learns how to gradually denoise from white Gaussian noises towards a realistic image. To train such a model, images are decayed stepwise with known parameters. The forward sampling process is expressed as

$$q(x_t|x_{t-1}) = \mathcal{N}(x_t; \sqrt{1-\beta_t}x_{t-1}, \beta_t\mathbf{I}) \tag{1}$$

Such sampling process continues for $T$ steps. A substitution

$$\bar{\alpha}_t = \prod_{i=1}^{t}(1-\beta_i) \tag{2}$$

enables efficient direct acquirement of the sampled image from original input at step $t$. That is,

$$q(x_t|x_0) = \mathcal{N}(x_t; \sqrt{\bar{\alpha}_t}x_0, (1-\bar{\alpha}_t)\beta_t\mathbf{I}) \tag{3}$$

The reverse sampling process

$$p(x_{t-1}|x_t) = \mathcal{N}(x_{t-1}; \mu_\theta(x_t, t), \Sigma_\theta(x_t, t)) \tag{4}$$

is to be learned. Ho et al. (2020) stated that the predicted mean can be expressed as

$$\mu_\theta(x_t, t) = \frac{1}{\sqrt{\alpha_t}}(x_t - \frac{\beta_t}{\sqrt{1-\bar{\alpha}_t}}\epsilon_\theta(x_t, t)) \tag{5}$$

The denoising function $\epsilon_\theta(x_t, t)$ is obtained through the minimization of objective function

$$L = E_{t,x_0,\epsilon}\|\epsilon - \epsilon_\theta(x_t, t)\|^2 \tag{6}$$

Using a u-net structure. We assume that there physically exists a "real" image that contains both the content from the untampered region and that tampered part.

On tampered image anti-forensics, Tailanián et al. (2024) proposes guided diffusion to purify tampered traces, adding a term $s_t$ in the denoising process in Equation (4) to control the extent of proximity between real and tampered image at the denoising step $t$.

$$p(x_{t-1}|x_t) = \mathcal{N}(x_{t-1}; \mu_\theta(x_t, t) - s_t\Sigma_\theta(x_t, t)\nabla_{x_t}\mathcal{D}(x_t, x_0), \Sigma_\theta(x_t, t)) \tag{7}$$

where $\mathcal{D}$ is a similarity measure between the input image and sample at step $t$. The guiding term $s_t$ is expressed as

$$s_t = s\frac{\sqrt{1-\bar{\alpha}_t}}{\sqrt{\bar{\alpha}_t}} \tag{8}$$

which is in negative correlation with step $t$, the rationale being greater guidance is needed at large $t$ for better content preservation, while at small $t$, guidance should be less for better forgery trace removal. However, the rich information contained in the realistic region is often overseen. Inspired by RePaint (Lugmayr et al. (2022)), which employs pretrained diffusion model and exploits existing regions to inpaint missing parts, we propose GUIDE to make full use of information in authentic regions of tampered images.

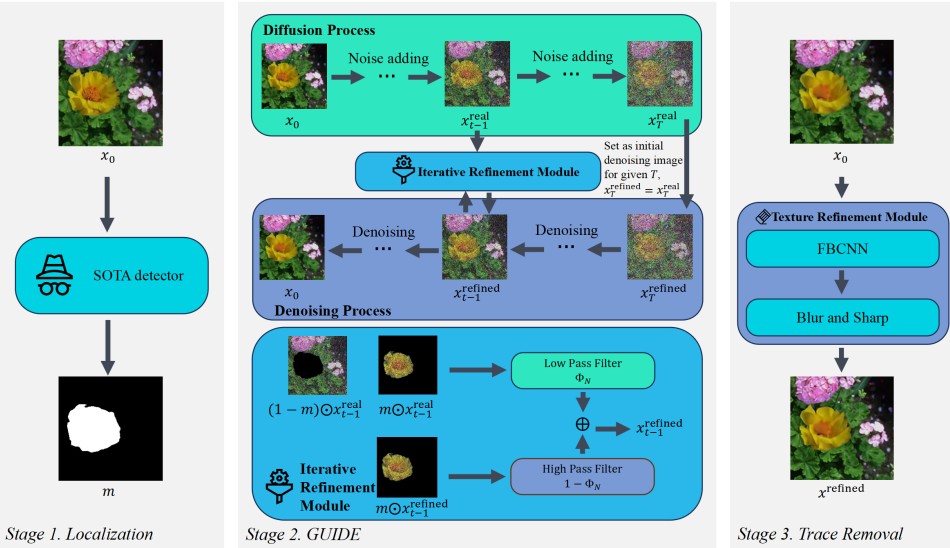

Figure 2: A schematic illustration of the proposed method is shown. Following the extraction of a localisation mask from the victim detector, an iterative diffusive refinement process is initiated, making full use of the authentic content delineated by the mask. Subsequently, the Texture Refinement Module (TRM) is employed to eliminate any residual artefacts.

## 3 APPROACH

To address the challenge of preserving the visual authenticity of tampered regions in existing anti-forensics algorithms, we propose a two-step refinement framework, as depicted in Fig. 2. Our approach involves two primary stages: generating realistic content using the GUIDE model and removing forgery artefacts via the Texture Refinement Module (TRM). Initially, the attacked detector produces a localisation map to identify the tampered regions. GUIDE then eliminates high-frequency artefacts, and TRM subsequently refines the image by addressing texture inconsistencies.

Rather than guiding the image towards the tampered image itself, which risks reverting the diffused image back into the tampered domain, in this paper, we rely entirely on the diffusion model's capability to generate authentic content by using only the low-frequency components as guidance.

**Mask obtaining**. We leverage information from the authentic regions to fix the tampered regions. First, a localisation mask $m$, generated by the victim detector $D$, is obtained to identify the tampered regions. In their study on perceptual artefact removal, Zhang et al. (2023) demonstrated that localising perceptual artefacts at a fine-grained level, rather than addressing the entire edited region, can improve performance. We hypothesise that forgery traces can be treated similarly. The use of this mask, instead of a ground truth mask, is based on the rationale that correctly identified regions contain the majority of artefacts, while misidentified areas highlight weaknesses in image forensics, offering additional information for refinement. Furthermore, we aim for the model to function effectively in scenarios where a ground truth mask is unavailable.

**Realistic guidance**. We initialise the denoising process from step $T$, which controls the extent of refinement in the manipulated region—the longer the time span, the more steps are taken to fuse authentic content with the manipulated area. In Equation (4), we modify the term $x_t$ in $p(x_{t-1}|x_t)$ during the reverse sampling process. At step $t$, we combine the authentic component with the diffusion-refined manipulated region as follows:

$$x_{t-1} = (1-m) \odot x_{t-1}^{\text{real}} + m \odot (LF(x_{t-1}^{\text{real}}) + HF(x_{t-1}^{\text{refined}})) \tag{9}$$

where $x_{t-1}^{\text{real}}$ denotes the sampled image from the input image, and $x_{t-1}^{\text{refined}}$ denotes the refined image from diffusion at step $t$. Specifically,

$$x_{t-1}^{\text{real}} = \sqrt{\bar{\alpha_{t-1}}}x_0 + (1 - \bar{\alpha_{t-1}})\epsilon \tag{10}$$

is derived from Equation (3) and

$$x_{t-1}^{\text{refined}} = \frac{1}{\sqrt{\alpha_t}}(x_t - \frac{\beta_t}{\sqrt{1 - \bar{\alpha_t}}}\epsilon_\theta(x_t, t)) + \sigma_t z \tag{11}$$

where $z \sim \mathcal{N}(\mathbf{0}, \mathbf{I})$ if $t > 1$, otherwise $z = \mathbf{0}$, derived from Equation (5). Through this process, we iteratively leverage information from realistic regions and utilise the low-frequency components as guidance to generate authentic high-frequency details.

The low-frequency component is obtained by:

$$LF(x_{t-1}^{\text{real}}) = \Phi_N(x_{t-1}^{\text{real}}) \tag{12}$$

where $\Phi_N(\cdot)$ denotes a low-pass filter controlled by the parameter $N$. A higher value of $N$ results in a lower boundary for the low-pass filter, thereby preserving less content (Wang et al. (2023)). The high-frequency component is then obtained by subtracting the low-frequency part from the image:

$$HF(x_{t-1}^{\text{refined}}) = (\mathbf{I} - \Phi_N)(x_{t-1}^{\text{refined}}) \tag{13}$$

**Resampling**. Inspired by RePaint (Lugmayr et al. (2022)), we periodically introduce the noise-adding process during denoising. This process is controlled by two hyper-parameters: jump length $j$, which denotes the frequency of resampling, and resampling times $u$, which governs the number of repetitions in one epoch of refinement. In this way, the authentic content of the tampered images is repeatedly sampled to assimilate the tampered regions, facilitating better recovery of harmonised high-frequency components. Forward mixing enables full exploitation of the real regions, resulting in improved performance for image anti-forensics. A scenario where $j = 1$ and $u = U$ is illustrated in Algorithm 1. Through iterative refinement, the algorithm ultimately produces a refined input $x_0$.

---

**Algorithm 1:** Guided Diffusive Refinement

**Input** : tampered image $x_0$, localization map generated by victim detector $m$
**Output:** refined image $x^{\text{refined}}$

1  $x_t \sim \mathcal{N}(\mathbf{0}, \mathbf{I})$;
2  **for** $t = T, ..., 1$ **do**
3      **for** $u = 1, ..., U$ **do**
4          $\epsilon \sim \mathcal{N}(\mathbf{0}, \mathbf{I})$ if $t > 1$, else $\epsilon = \mathbf{0}$
5          $x_{t-1}^{\text{real}} = \sqrt{\bar{\alpha}_t}x_0 + (1 - \bar{\alpha}_t)\epsilon$
6          $LF(x_{t-1}^{\text{real}}) = \Phi_N(x_{t-1}^{\text{real}})$
7          $z \sim \mathcal{N}(\mathbf{0}, \mathbf{I})$ if $t > 1$, else $z = \mathbf{0}$
8          $x_{t-1}^{\text{refined}} = \frac{1}{\sqrt{\alpha_t}}(x_t - \frac{\beta_t}{\sqrt{1-\bar{\alpha_t}}}\epsilon_\theta(x_t, t)) + \sigma_t z$
9          $HF(x_{t-1}^{\text{refined}}) = (\mathbf{I} - \Phi_N)(x_{t-1}^{\text{refined}})$
           $x_{t-1} = (1 - m) \odot x_{t-1}^{\text{real}} + m \odot (LF(x_{t-1}^{\text{real}}) + HF(x_{t-1}^{\text{refined}}))$
10     **end**
11     **if** $u < U$ *and* $t > 1$ **then**
12         $x_t \sim \mathcal{N}(\sqrt{1 - \beta_{t-1}}x_{t-1}, \beta_{t-1}\mathbf{I})$
13     **end**
14 **end**
15 $x^{\text{refined}} = f_{\text{TRM}}(x_0)$
16 **return** $x^{\text{refined}}$

---

**Texture Refinement Module.** As authentic traces are added, tampered trace removal is applied to address the possibility that the diffusion model may not completely eliminate camera-specific noiseprints. This limitation arises because the pretrained diffusion model, based on ImageNet, may not specialise in generating equipment-specific details. However, such traces are heavily exploited

by detectors like TruFor (Guillaro et al. (2023)). To mitigate this, we incorporate the Texture Refinement Module (TRM) to erase artefacts such as noise patterns, which enhances the overall performance of our model:

$$x^{\text{refined}} = f_{TRM}(x_0) \tag{14}$$

Specifically, we combined FBCNN and Blur & Sharp method to form the Texture Refinement Module (TRM). The JPEG artefact removal method FBCNN (Jiang et al. (2021)) is capable of improving image quality and has demonstrated impressive performance in evading noise-based detection methods. Meanwhile, the Blur & Sharp method smooth local textures by applying a custom-designed Gaussian blurring kernel and sharpening kernel across the entire image. As described in Tahir & Bal (2024), the blurring kernel $A$ and sharpening kernel $B$ are defined as follows:

$$A = \begin{pmatrix} 1 & 4 & 7 & 4 & 1 \\ 4 & 16 & 26 & 16 & 4 \\ 7 & 26 & 41 & 26 & 7 \\ 4 & 16 & 26 & 16 & 4 \\ 1 & 4 & 7 & 4 & 1 \end{pmatrix}, \ B = \begin{pmatrix} 0 & -1 & 0 \\ -1 & 5 & -1 \\ 0 & -1 & 0 \end{pmatrix},$$

The Blur & Sharp process eliminate abrupt peaks and troughs within the image, further harmonising the overall textures. We conducted extensive experiments to evaluate whether the two-step refinement scheme provides satisfactory results. The effectiveness of our pipeline is rooted in its integration of prior image forensics methods: the construction of authentic traces using GUIDE and the removal of detectable traces through TRM.

## 4 EXPERIMENTS

**Victim Detector.** We selected six representative forensic methods for our evaluation. The selected forensic techniques include: SPAN (Hu et al. (2020)), MVSS-Net (Chen et al. (2021)), IF-OSN (Wu et al. (2022)), TruFor (Guillaro et al. (2023)), MMFusion-IML (Triaridis & Mezaris (2024)), and EITLNet (Guo et al. (2024))—each exploiting different types of forgery traces, to evaluate the performance of our model. We utilised AUC and F1 scores as localisation metrics, where lower values indicate better anti-forensics performance. The taxonomy of the victim detectors is provided in Table 1 of the supplementary material.

**Dataset and Comparison Methods.** Experiments were conducted on three datasets: CASIAv2 (Dong et al. (2013)), COVERAGE (Wen et al. (2016)), and IMD2020 (Novozamsky et al. (2020)). Additional anti-forensics methods for comparison include *Diff-cf* and *Diff-cfg* (Tailanián et al. (2024)), FBCNN (Jiang et al. (2021)), Blur & Sharp, and Downsize & Upsize (Tahir & Bal (2024)).

**Implementation Details.** For the application of the pre-trained $256 \times 256$ diffusion model, we centre-cropped all manipulated images and moved those with cropped tampered areas to the authentic test set. We set a default jump length of $j = 10$ and resampling times of $u = 10$, as adopted by RePaint. The GUIDE model was executed on eight NVIDIA GeForce RTX 4090 GPUs. Given the image size constraint of $256 \times 256$, the low-pass filter factor $N$ can only take integer factors that exactly divide 256, such as 2, 4, 8, 16, 32, etc. We selected a low-pass filter factor of $N = 8$ to reconstruct the maximum amount of detail while preserving the semantic content within the image. Further rationale for the selection of the time step $T$ and filter factor $N$ can be found in Section 6.

### 4.1 RESULTS

**Anti-forensics performance.** We analyse the image forensics performance as presented in Table 1. When considering the diffusion-based refinement module alone, GUIDE demonstrates superior performance compared to both *Diff-cf* and *Diff-cfg* in most scenarios, highlighting the effectiveness of authentic content guidance over complete image guidance. Fig. 3 showcases GUIDE's ability to generate authentic high-frequency components within manipulated images. Additionally, Fig. 2 in supplementary material illustrates that GUIDE produces authentic details that are consistent with the utilised authentic content, further supporting its efficacy in image anti-forensics.

Table 1: Performance comparison of the proposed method and other approaches across different datasets and forensic methods. Best is marked with red and second best is marked with blue. Lower metric values indicate better performance for anti-forensics.

| | | | Image Forensics Metrics(▼) | | | | | | | | | | | |
|---|---|---|---|---|---|---|---|---|---|---|---|---|---|---|
| | | detector | TruFor | | MVSS-Net | | IF-OSN | | SPAN | | MMFusion-IML | | EITLNet | |
| dataset | | Anti-forensics | AUC | F1 | AUC | F1 | AUC | F1 | AUC | F1 | AUC | F1 | AUC | F1 |
| CASIAv2 | | Original | 0.8828 | 0.9798 | 0.8244 | 0.7209 | 0.8643 | 0.4812 | 0.7966 | 0.2739 | 0.8875 | 0.7358 | 0.7768 | 0.4217 |
| | Others | Diff-cf | 0.8367 | 0.9716 | 0.7905 | 0.6978 | 0.8200 | 0.3872 | 0.7635 | 0.2414 | 0.8393 | 0.6266 | 0.6848 | 0.2383 |
| | | Diff-cfg | 0.8453 | 0.9710 | 0.7886 | 0.7017 | 0.8200 | 0.3872 | 0.7635 | 0.2414 | 0.8526 | 0.7827 | 0.6950 | 0.2533 |
| | | FBCNN | 0.7952 | 0.9520 | 0.8830 | 0.7490 | 0.7376 | 0.2936 | 0.7338 | 0.2202 | 0.8177 | 0.5517 | 0.7123 | 0.3343 |
| | | Downsize & Upsize | 0.5800 | 0.9697 | 0.8159 | 0.7121 | 0.8320 | 0.3964 | 0.6371 | 0.1810 | 0.7031 | 0.5258 | 0.8214 | 0.4480 |
| | | Blur & Sharp | 0.5427 | 0.9644 | 0.8585 | 0.7359 | 0.8299 | 0.3857 | 0.6816 | 0.1820 | 0.6042 | 0.4655 | 0.7996 | 0.4204 |
| | Ours | TRM | 0.5653 | 0.9546 | 0.8408 | 0.7312 | 0.7553 | 0.2773 | 0.6748 | 0.1811 | 0.5621 | 0.3446 | 0.7432 | 0.2665 |
| | | GUIDE, T=250 | 0.8082 | 0.9727 | 0.7557 | 0.6685 | 0.8621 | 0.4459 | 0.7381 | 0.2053 | 0.8454 | 0.6425 | 0.7553 | 0.3775 |
| | | GUIDE, T=1000 | 0.7796 | 0.9692 | 0.7669 | 0.6863 | 0.8432 | 0.4121 | 0.7473 | 0.2131 | 0.8132 | 0.5967 | 0.7371 | 0.3476 |
| | | GUIDE+TRM, T=250 | 0.6008 | 0.9572 | 0.8108 | 0.7103 | 0.7651 | 0.2721 | 0.6907 | 0.1815 | 0.6069 | 0.3901 | 0.7672 | 0.2797 |
| | | GUIDE+TRM, T=1000 | 0.5717 | 0.9552 | 0.8238 | 0.7195 | 0.7492 | 0.2589 | 0.6783 | 0.1813 | 0.5668 | 0.3395 | 0.7631 | 0.2782 |
| COVERAGE | | Original | 0.6747 | 0.9362 | 0.7337 | 0.2529 | 0.7147 | 0.1256 | 0.7958 | 0.2769 | 0.5398 | 0.3241 | 0.7409 | 0.2584 |
| | Others | Diff-cf | 0.6164 | 0.9363 | 0.5808 | 0.2521 | 0.6913 | 0.1083 | 0.7086 | 0.2349 | 0.5398 | 0.3241 | 0.7409 | 0.2584 |
| | | Diff-cfg | 0.6687 | 0.9297 | 0.6376 | 0.2510 | 0.6955 | 0.1293 | 0.7164 | 0.2419 | 0.4864 | 0.2680 | 0.7225 | 0.2258 |
| | | FBCNN | 0.6239 | 0.9264 | 0.5642 | 0.2493 | 0.6963 | 0.1638 | 0.6961 | 0.2302 | 0.5143 | 0.2465 | 0.7029 | 0.2326 |
| | | Downsize & Upsize | 0.6193 | 0.9289 | 0.6450 | 0.2517 | 0.7267 | 0.1615 | 0.6443 | 0.2233 | 0.5264 | 0.2574 | 0.6868 | 0.1028 |
| | | Blur & Sharp | 0.6757 | 0.9332 | 0.5873 | 0.2506 | 0.7264 | 0.1323 | 0.6825 | 0.2220 | 0.5047 | 0.2608 | 0.7132 | 0.1489 |
| | Ours | TRM | 0.6811 | 0.9323 | 0.4588 | 0.2446 | 0.7191 | 0.1322 | 0.6740 | 0.2234 | 0.5410 | 0.2327 | 0.7046 | 0.1541 |
| | | GUIDE, T=250 | 0.6718 | 0.9377 | 0.7151 | 0.2521 | 0.7114 | 0.1287 | 0.7681 | 0.2443 | 0.5181 | 0.2991 | 0.7293 | 0.2457 |
| | | GUIDE, T=1000 | 0.6699 | 0.9352 | 0.7239 | 0.2529 | 0.7064 | 0.1487 | 0.7633 | 0.2468 | 0.5029 | 0.2985 | 0.7178 | 0.2504 |
| | | GUIDE+TRM, T=250 | 0.6871 | 0.9358 | 0.4269 | 0.2424 | 0.7225 | 0.1191 | 0.6868 | 0.2235 | 0.5180 | 0.2399 | 0.7014 | 0.1535 |
| | | GUIDE+TRM, T=1000 | 0.6762 | 0.9367 | 0.4431 | 0.2424 | 0.7248 | 0.1292 | 0.6865 | 0.2235 | 0.5201 | 0.2441 | 0.7025 | 0.1553 |
| IMD2020 | | Original | 0.7261 | 0.9021 | 0.6708 | 0.5129 | 0.8030 | 0.4001 | 0.7282 | 0.3712 | 0.8591 | 0.7202 | 0.7845 | 0.4780 |
| | Others | Diff-cf | 0.5681 | 0.8833 | 0.5671 | 0.4837 | 0.7446 | 0.2892 | 0.6991 | 0.3266 | 0.7589 | 0.6198 | 0.6739 | 0.2851 |
| | | Diff-cfg | 0.6296 | 0.8928 | 0.5751 | 0.4864 | 0.7538 | 0.3319 | 0.6935 | 0.3283 | 0.8090 | 0.6492 | 0.6901 | 0.3731 |
| | | FBCNN | 0.6381 | 0.8792 | 0.6879 | 0.5187 | 0.7041 | 0.2986 | 0.6880 | 0.3258 | 0.8001 | 0.6053 | 0.6822 | 0.3748 |
| | | Downsize & Upsize | 0.6034 | 0.8915 | 0.6153 | 0.5000 | 0.7707 | 0.3060 | 0.6396 | 0.3201 | 0.8088 | 0.6627 | 0.7628 | 0.4189 |
| | | Blur & Sharp | 0.5138 | 0.8840 | 0.6381 | 0.5105 | 0.7802 | 0.2854 | 0.7165 | 0.3212 | 0.7328 | 0.6243 | 0.7567 | 0.3817 |
| | Ours | TRM | 0.5430 | 0.8688 | 0.5217 | 0.4730 | 0.7210 | 0.2390 | 0.7110 | 0.3218 | 0.7301 | 0.5378 | 0.6946 | 0.2646 |
| | | GUIDE, T=250 | 0.7009 | 0.9000 | 0.5546 | 0.4783 | 0.8160 | 0.4040 | 0.6965 | 0.3249 | 0.8376 | 0.6677 | 0.8003 | 0.4747 |
| | | GUIDE, T=1000 | 0.6389 | 0.8912 | 0.5890 | 0.4914 | 0.7948 | 0.3804 | 0.6949 | 0.3279 | 0.8126 | 0.6539 | 0.7666 | 0.4253 |
| | | GUIDE+TRM, T=250 | 0.5415 | 0.8691 | 0.4514 | 0.4428 | 0.7285 | 0.2332 | 0.6595 | 0.2200 | 0.7218 | 0.5218 | 0.6595 | 0.2200 |
| | | GUIDE+TRM, T=1000 | 0.5321 | 0.8676 | 0.4698 | 0.4491 | 0.7156 | 0.2329 | 0.6766 | 0.2447 | 0.7215 | 0.5190 | 0.6766 | 0.2447 |

We further evaluate the performance of the TRM module. Although it does not achieve the highest performance individually, TRM exhibits balanced anti-forensics capabilities across all detectors when compared to other anti-forensics methods, providing a significant boost to GUIDE. Overall, the combination of GUIDE and TRM performed best on the IMD2020 dataset, achieving state-of-the-art results across nearly all metrics and detectors. Notably, we observe complementary effects between GUIDE and TRM, particularly with SPAN, where the F1 score of IF-OSN experienced a sharp decline of over $10\%$ compared to when GUIDE or TRM was used alone. As illustrated in Fig. 2 in the supplementary material, the combination of GUIDE and TRM effectively deceives detectors, almost completely eliminating identifiable forgery traces in the selected images.

While our method demonstrates significant improvements in many cases, there are instances where it does not achieve state-of-the-art performance. Part of this ineffectiveness can be attributed to the varying attention areas across different detectors; not all detectors produce the same localisation results as TruFor, which leads to GUIDE refining only limited regions. Additionally, when the manipulated area is particularly large, the available authentic information may be insufficient for GUIDE to completely refine the image.

**Image quality performance.** As shown in Table 2, *Diff-cf* yields superior non-reference image quality results compared to GUIDE, as it employs fewer steps in its diffusion process, thereby maintaining a higher level of harmony. In contrast, GUIDE achieves near-optimal non-reference metrics and performs well on reference-quality results, demonstrating strong quality and content preservation. This effectiveness is attributed to GUIDE's direct utilisation of authentic content from the tampered image while refining only the identified manipulated areas.

Meanwhile, the Blur & Sharp, Downsize & Upsize methods result in a greater loss of image content. This occurs because these two operations function similarly to a universal averaging unit that

Table 2: Image Quality Assessments. Best is marked with red and second best is marked with blue.

| | | Image Quality Metrics | | | | |
| --- | --- | --- | --- | --- | --- | --- |
| | metric type | None reference | | Reference | | |
| dataset | manipulators | BRISQUE(▼) | NIQE(▼) | PSNR(▲) | SSIM(▲) | LPIPS(▼) |
| CASIAv2 | Original | 26.576 | 7.1071 | - | - | - |
| | Diff-cf | 25.157 | 6.9604 | 31.55 | 0.8934 | 0.0954 |
| | Diff-cfg | 27.454 | 7.6469 | 32.03 | 0.9200 | 0.0790 |
| | FBCNN | 28.960 | 7.9883 | 32.43 | 0.9207 | 0.1063 |
| | Downsize & Upsize | 35.716 | 8.9123 | 32.34 | 0.8837 | 0.1340 |
| | Blur & Sharp | 37.664 | 8.6606 | 32.29 | 0.9026 | 0.1252 |
| | TRM | 37.664 | 8.6617 | 31.36 | 0.8596 | 0.1555 |
| | GUIDE, T=250 | 25.384 | 6.9892 | 40.64 | 0.9413 | 0.0422 |
| | GUIDE, T=1000 | 25.481 | 6.8946 | 40.60 | 0.9396 | 0.0398 |
| | GUIDE+TRM, T=250 | 37.822 | 8.5121 | 31.14 | 0.8150 | 0.1813 |
| | GUIDE+TRM, T =1000 | 37.734 | 8.5404 | 31.13 | 0.8140 | 0.1793 |
| COVERAGE | Original | 22.191 | 6.2504 | - | - | - |
| | Diff-cf | 19.919 | 6.0650 | 34.24 | 0.9183 | 0.0758 |
| | Diff-cfg | 27.436 | 6.7780 | 35.16 | 0.9449 | 0.0611 |
| | FBCNN | 36.989 | 7.9773 | 36.13 | 0.9541 | 0.0721 |
| | Downsize & Upsize | 35.104 | 6.6755 | 34.98 | 0.9413 | 0.0909 |
| | Blur & Sharp | 40.625 | 8.7229 | 34.41 | 0.9406 | 0.0806 |
| | TRM | 42.089 | 9.2147 | 33.50 | 0.9143 | 0.1060 |
| | GUIDE, T=250 | 24.596 | 6.4091 | 43.76 | 0.9751 | 0.0226 |
| | GUIDE, T=1000 | 24.199 | 6.3252 | 43.79 | 0.9742 | 0.0211 |
| | GUIDE+TRM, T=250 | 42.061 | 9.3113 | 33.29 | 0.8984 | 0.1193 |
| | GUIDE+TRM, T=1000 | 42.035 | 9.2313 | 33.29 | 0.8976 | 0.1178 |
| IMD2020 | Original | 21.972 | 5.6132 | - | - | - |
| | Diff-cf | 18.830 | 5.5026 | 33.73 | 0.8867 | 0.1215 |
| | Diff-cfg | 24.208 | 5.8482 | 34.48 | 0.9167 | 0.0996 |
| | FBCNN | 33.006 | 7.0256 | 36.07 | 0.9383 | 0.1012 |
| | Downsize & Upsize | 33.389 | 6.6117 | 37.21 | 0.9478 | 0.0743 |
| | Blur & Sharp | 32.226 | 7.7208 | 36.48 | 0.9468 | 0.0775 |
| | TRM | 34.258 | 8.2967 | 34.61 | 0.9085 | 0.1130 |
| | GUIDE, T=250 | 24.272 | 5.9897 | 40.13 | 0.9369 | 0.0644 |
| | GUIDE, T=1000 | 23.334 | 5.7986 | 40.05 | 0.9339 | 0.0624 |
| | GUIDE+TRM, T=250 | 35.073 | 8.5930 | 33.78 | 0.8618 | 0.1554 |
| | GUIDE+TRM, T=1000 | 34.978 | 8.4760 | 33.75 | 0.8594 | 0.1539 |

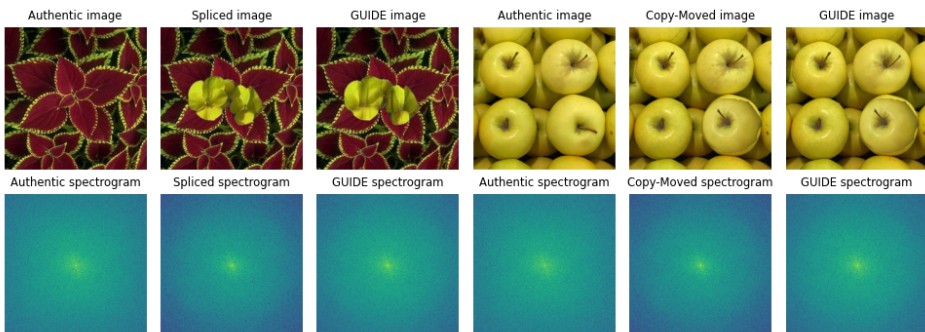

Figure 3: Spectrograms of authentic image, tampered image, and GUIDE images. Manipulated images leaves high frequency artefacts, while GUIDE has strong capability of constructing authentic high frequency component for manipulated images.

lowers image resolution, significantly modifying individual pixel values. While the overall structure remains intact, local details experience considerable degradation. As illustrated in Fig. 3 of the supplementary material, these two methods cause the images to appear blurrier. Consequently, although forgery traces are removed more effectively with the combination of GUIDE and TRM, this comes at the cost of a more pronounced loss in image quality compared to GUIDE used alone.

## 4.2 ABLATION STUDY

In this section, we selected a sample of 108 images from the IMD2020 dataset to investigate how the number of steps $T$ affects the anti-forensics performance of GUIDE+TRM. As illustrated in Fig. 4, $T$ has varying effects across different detectors.

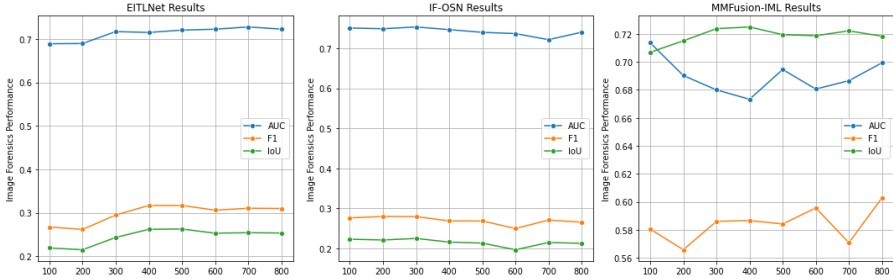

Figure 4: Comparison of anti-forensics performance on different detectors regarding different $T$. Best performance is spotted at different initiation steps $T$, 200, 600 and 200, respectively.

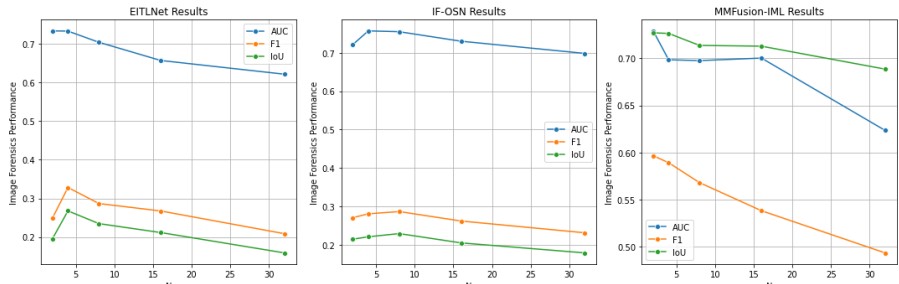

Figure 5: Comparison of anti-forensics performance on different detectors regarding different $N$. Overall, less low-frequency content kept enables a greater amount of high-frequency detail generated, leading to more effective anti-forensics.

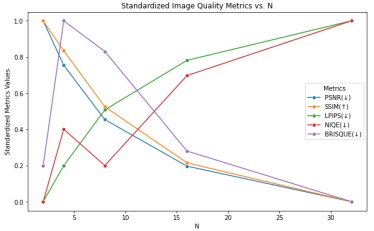

Figure 6: Standardized image quality metrics at different $N$.

We compared AUC, F1, and IoU metrics for EITLNet, IF-OSN, and MMFusion-IML. The results demonstrate that the relationship between the metrics and $T$ is not monotonous but rather dynamically changing. In the selected sample, the best overall performance on EITL-Net is observed at $T = 200$, while for IF-OSN, it is at $T = 800$, and for MMFusion-IML, it is at $T = 200$. This indicates that the extent of GUIDE refinement produces different complementary effects with TRM across various image forensics methods. Thus, we select a shorter $T = 250$ and longer $T = 1000$ for a more comprehensive experiment. Additionally, Fig. 5 reveals that maintaining less of the original low-frequency content leads to improved anti-forensics performance. However, as shown in Fig.6, this reduction results in a significant loss of image quality due to decreased semantic information. Therefore, to strike a balance between preserving semantic content and optimising image forensics performance, we select $N = 8$.

## 5 CONCLUSION

In this paper, we presented a novel two-stage approach to image anti-forensics, addressing the challenges posed by high-frequency artefacts in manipulated images. Our method, GUIDE, leverages zero-shot learning through diffusion-based refinement, effectively restoring details by exploiting low-frequency information from authentic regions. Additionally, we introduced a texture refinement module to remove residual artefacts, enhancing the overall anti-forensics performance. Extensive experiments on multiple forensic datasets confirm the effectiveness of our approach, surpassing existing methods, particularly in terms of balancing forgery trace removal with content preservation. Our findings demonstrate the potential of diffusion models to advance the field of image anti-forensics, offering a robust solution for evading forensic detectors while maintaining visual authenticity.

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
