# A APPENDIX

Table 1: Taxonomy of victim detectors

| Methods | Views | | | Backbone | scales of supervision | | |
| --- | --- | --- | --- | --- | --- | --- | --- |
| | RGB | Noise | Fusion | | pixel | edge | image |
| SPAN (Hu et al. (2020)) | + | SRM filter, BayarConv2D | early fusion (feature concatenation) | WiderVGG | + | - | - |
| MVSS-Net (Chen et al. (2021)) | + | BayarConv2D | late fusion (dual attention) | FCN | + | + | + |
| IF-OSN (Wu et al. (2022)) | - | SE-U-Net | - | DNN | + | - | - |
| TruFor (Guillaro et al. (2023)) | + | DnCNN | early fusion (cross modal fusion) | Transformer | + | - | - |
| MMFusion-IML (Triaridis & Mezaris (2024)) | + | SRM filter, BayarConv2D, Noiseprint++ | early fusion (cross modal fusion) | CNN | + | - | - |
| EITLNet (Guo et al. (2024)) | + | CW-HPF | late fusion (feature concatenation) | Transformer | + | - | - |

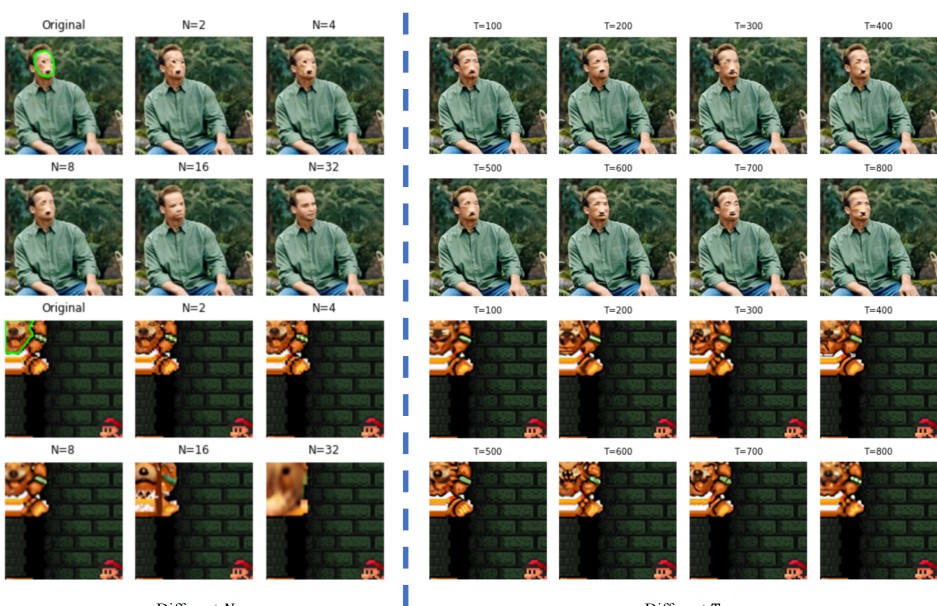

Figure 1: Illustration of effects of different $N$ and $T$ on image content. Different $N$ and $T$ result in different refined image contents. Overall, greater $N$ leads to less original content preserved, while larger $T$ generates more harmonious images.

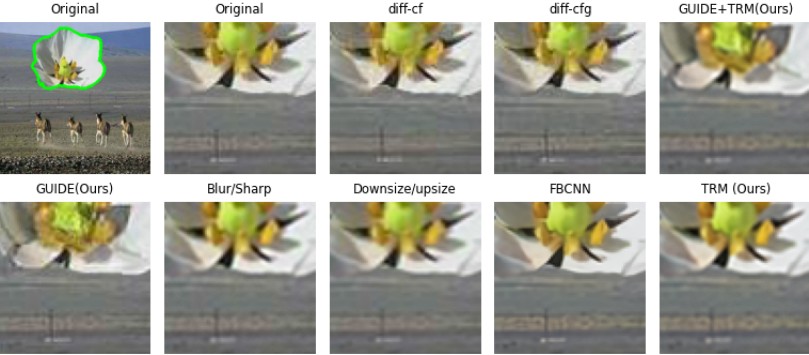

Figure 2: Texture and quality comparisons between different anti-forensics methods. GUIDE generates content-specific details compatible with authentic regions.

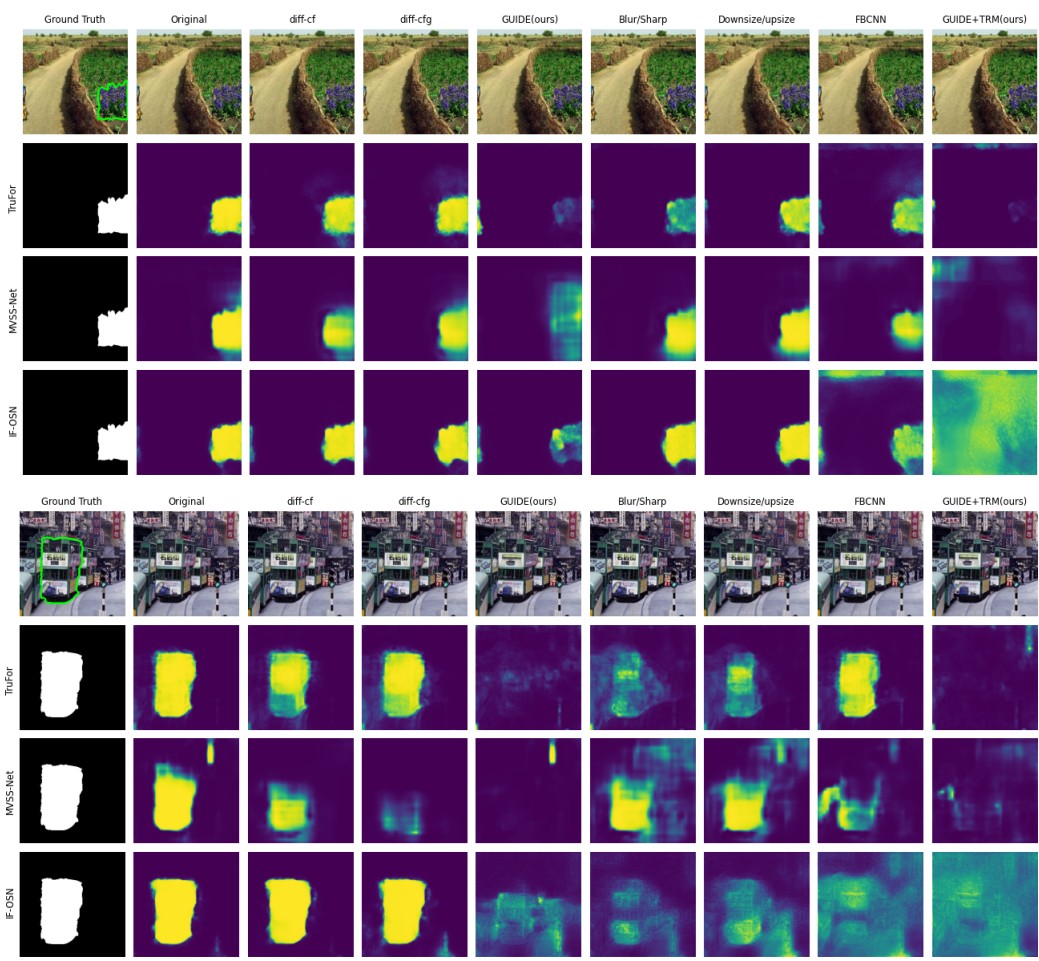

Figure 3: Comparison of anti-forensics performance on different detectors with their localization maps. As compared to other diffusion based method, *Diff-cf* and *Diff-cfg*, GUIDE more effectively fools image anti-forensics, while TRM enhances such performance.