# OpenReview forum: "From Forgery to Authenticity: Image Anti-Forensics via Reconstruction and Artefact Elimination"
_ICLR.cc/2025/Conference — ICLR 2025 Conference Withdrawn Submission_

### Official Review · Reviewer_bJvb · 2024-10-28

**Soundness:** 2
**Presentation:** 3
**Contribution:** 2
**Rating:** 3
**Confidence:** 4

**Summary:**

Image forensics, or more specifically, image manipulation detection, is a common technique used to detect image tampering methods such as Photoshop.

This paper proposes an attack method targeting automated image forensics techniques, which can be summarized as an anti-forensic approach.

The goal is to confuse existing SoTA methods in accurately identifying tampered images. Specifically, this paper proposes a two-stage method: in the first stage, Diffusion is used to refine the details of the tampered regions based on real image areas. In the second stage, performance is further adjusted by applying refinement filters to enhance texture.

**Strengths:**

- Anti-forensics is an emerging field with very limited related research, currently only one or two published studies. Existing research focuses more on anti-deepfake detection; however, there are differences in datasets and evaluation models.
- "The method proposed in this paper performs well in both confusing existing Image-Forensic models and maintaining the quality of the original image through experiments.
- The model design is very intuitive, and the motivation is clear.

**Weaknesses:**

- Some implementation details and protocols provided in the paper is unclear
  - The term 'localization' is mentioned multiple times in the paper, but in mainstream research contexts, such as TruFor [(Guillaro et al., 2023)], 'detection' is generally considered as image classification (image-level), and 'localization' as image segmentation (pixel-level). However, the metrics in Table 1 are extremely high, so I'm not entirely sure whether these are the results of detection or localization; the paper does not make this clear.
  - Additionally, the description of the specific datasets and how the experiments were conducted is unclear. For instance, there is no information on which dataset was used for training, which for testing, or whether real images were included, etc. (lines 357–360).

- The structure of Stage 2 is relatively simple and is derived from two previous works, leading to a lack of overall technical contributions in this paper.

- Why aren’t TruFor and MVSS-net included as comparisons in Figures 4 and 5? Further, I kindly suggest changing these tables to separate a figure for each AUC, F1, and IOU, each figure contains five curves representing each model. The current figure putting AUC, F1, and IOU on the same chart is unreasonable.

**Questions:**

I have a minor doubt whether there is sufficient necessity to conduct research in this field:
- Currently, Image Forensics mostly remains at the experimental stage, with few commercial applications. Similarly, adversarial attack techniques on object detection only began to be published at top conferences once these technologies were widely adopted in society[A]. Researching attack techniques on image forensics may be **somewhat premature** at this moment.


## Reference
[A] Nguyen, K. N. T., Zhang, W., Lu, K., Wu, Y., Zheng, X., Tan, H. L., & Zhen, L. (2024). A Survey and Evaluation of Adversarial Attacks for Object Detection. arXiv preprint arXiv:2408.01934.

---

### Official Review · Reviewer_5LdP · 2024-10-29

**Soundness:** 2
**Presentation:** 1
**Contribution:** 2
**Rating:** 3
**Confidence:** 4

**Summary:**

The authors propose a two-phase approach. In the first phase, they introduce GUIded Diffusive rEfinement(GUIDE). In the second phase, we introduce anartefact removal algorithm. Their contributions are summarised as follows:

(1) They introduce a zero-shot diffusion-based refinement method that fully exploits the information from authentic regions of tampered images.

(2) They propose a two-stage refinement framework that achieves state-of-the-art performance across various forensic detection benchmarks.

(3) They conduct an in-depth exploration of the trade-off between effective image anti-forensics and overall image quality.

**Strengths:**

Existing image anti-forensics methods primarily focus on rectifying artefacts at the feature level, often overlooking the authenticity of the manipulated regions. The authors propose a novel two-stage image anti-forensics framework that leverages guided diffusion refinement to address the limitations of existing methods.

**Weaknesses:**

(1) I don't understand what the author is trying to say at each step in Figure 1. For example, what is the green dot and why does it disappear? Does the decision boundary of detector A refer to the curve in the center or the green dot? The authors did not express their work well with Figure 1.

(2) Authors should not spend nearly 2 pages in a conference article describing related work. Authors should focus on the presentation of the proposed methodology.

(3) I also can't understand the meaning expressed by Stage 2 in Fig. 2. Does the low pass filter in the bottom refer to the diffusion process at the top? If so, why is the diffusion process called a low pass filter? If not, what is the relationship between the top box and the bottom box?

(4) As shown in Figure 2, the author's method relies heavily on the effect of the initial detector. Does this mean that if the detector cannot locate the location of the tampered area, the author's method fails?

**Questions:**

The author needs to explain in detail the questions I mentioned above, and I will determine the final score based on your answers.

---

### Official Review · Reviewer_nfJ4 · 2024-10-31

**Soundness:** 1
**Presentation:** 2
**Contribution:** 2
**Rating:** 3
**Confidence:** 4

**Summary:**

This work presents an image anti-forensics method to evade detection from image forgery detectors. The method focuses on refining manipulated images both in terms of artefact removal and authenticity in altered regions. The two-phase approach consists of:

1. GUIDE (GUIded Diffusive rEfinement): A zero-shot learning-based module for reconstructing details in unaltered areas of the image.

2. Artefact Removal: A targeted algorithm to clean up artefacts specifically in the forged areas of the image.

The experimental results on several datasets and detectors shows the effectiveness of the designed anti-forensics framework.

**Strengths:**

This work represents an early attempt to perform image anti-forensics, paving the way for future research in this area.

**Weaknesses:**

1. The motivation for developing an anti-forensics framework is not adequately discussed in the introduction. The authors should provide a thorough analysis and clear justification of the research purpose and motivations.

2. At the end of the introduction, the authors mention analyzing the trade-off between anti-forensics performance and image quality. However, intuitively, higher image quality should enhance anti-forensics effectiveness from a human perspective. This should be better substantiated in the manuscript.

3. The authors elaborate on the detailed diffusion process in Section 2.3, but this is not particularly informative and should not be included in the related works section.

4. The novelty of the proposed framework appears limited. Although the authors combine a diffusion-based module with a traditional texture refinement module to create anti-forensics images, the motivation behind this combination remains unclear.

5. As shown in Table 1, the proposed method's experimental results are suboptimal across different datasets and detectors. Additional datasets, such as Columbia, CASIAv1, and Carvalho, should be included to make the experiments more comprehensive.

6. This work does not include state-of-the-art (SOTA) detectors. The latest detection model used in experiments is Trufor, introduced at CVPR 2023.

**Questions:**

In Eq. 9, does the add operation mean pixel-level addition?

---

### Official Review · Reviewer_b7ML · 2024-10-31

**Soundness:** 3
**Presentation:** 3
**Contribution:** 2
**Rating:** 5
**Confidence:** 4

**Summary:**

This work presents a two-stage image anti-forensics framework. In the first stage,  GUIDE is proposed to enhance the authenticity of tampered regions by exploiting information from authentic areas. In the second stage, TRM is proposed to further eliminate the tampered traces.

**Strengths:**

This paper is easy to follow.

Exploiting information from authentic areas to improve anti-forensics performances is an interesting idea.

**Weaknesses:**

Although the idea is interesting, the contribution of this work is generally incremental. The framework of GUIDE is similar to RePaint \[1]. By comparing the **Algorithms part** between GUIDE and RePaint, the primary modification seems to be adding a frequency module. And TRM only comprises a prior FBCNN and the common Blur & Sharp operation.

To convincingly demonstrate the effectiveness of this contribution, it would be better to add a comparative analysis with the original RePaint and the proposed GUIDE, such as implementing RePaint + TRM settings for comparison.\\

\[1] RePaint: Inpainting using denoising diffusion probabilistic models. CVPR 2022

**Questions:**

For the image quality performances (Table 2), the setting of the **Reference Metric** is " authentic v.s. anti-tampered image" or "tampered vs anti-tampered image"?

---

### Official Review · Reviewer_4xer · 2024-11-09

**Soundness:** 3
**Presentation:** 2
**Contribution:** 3
**Rating:** 3
**Confidence:** 5

**Summary:**

This paper proposes a method towards image anti-forensics where the goal is to ensure that digitally manipulated images evade image forensics detection algorithms. The proposed method is based on a two-phase approach, where in the first phase, a diffusion based method is used to reconstruct details from unaltered regions, and in the second phase, an artifact removal algorithm removes artifacts from reconstructed manipulated regions. Experiments are presented on many datasets to validate the approach.

**Strengths:**

The paper targets an important problem of image antiforensics where the goal is to hide traces such that it will be difficult for image forensics algorithms to reliably detect image forgeries. The problem itself is important and is usually less addressed in academic studies.

The two-phase approach is presented well. The experiments are also detailed well. It is good that the paper also included visual examples.

**Weaknesses:**

In Figure 1, page 4 and Section 2.3, there are several equations which start with x and other symbols. However, many of these symbols do not seem to be defined anywhere in the paper. Some of them are defined but many are not. These are basic requirements in a technical paper and need to be addressed.

From the paper, it appears that the proposed approach build upon Diff-cfg (as mentioned inn the paper). It is not fully clear how much better the proposed approach is in comparison? It will be helpful to have a section which demonstrates what exactly is different in this paper and how it helps the overall performance. Though some of these details are discussed, it is not fully obvious from the paper.

Table 1 and 2 are crowded. It is difficult to see and infer how the proposed approach is actually performing in comparison to others. It is better to split these tables and show only the best cases of other detectors and show how the proposed approach compares to those. The detailed results could be documented in the supplementary section.

**Questions:**

What are the limitations of the proposed approach?

If an adversary is aware of the anti-forensics, can the adversary take measures to make the images similar to authentic images (just like how the proposed approach is doing)?

---

### Note · Authors · 2024-11-13

I have read and agree with the venue's withdrawal policy on behalf of myself and my co-authors.